# Structured matching models in multimodal information fusion: An optimized Kuhn-Munkres algorithm

Qingnan Ji[1,2], Jinxia Wang [ID][2]*, Lixian Wang[1]

1 Shaanxi University of International Trade and Commerce, Xi'an, China, 2 Art design College of Shaanxi Fashion Engineering University, Xi'an, China

* 1754354519@qq.com

## Abstract

In modern multimodal interaction design, integrating information from diverse modalities—such as speech, vision, and text—presents a significant challenge. These modalities differ in structure, timing, and data volume, often leading to mismatches, low computational efficiency, and suboptimal user experiences during the integration process. This study aims to enhance both the efficiency and accuracy of multimodal information fusion. To achieve this, publicly available datasets—Carnegie Mellon University Multimodal Opinion Sentiment Intensity (CMU-MOSI) and Interactive Emotional Dyadic Motion Capture (IEMOCAP)—are employed to collect speech, visual, and textual data relevant to multimodal interaction scenarios. The data undergo preprocessing steps including noise reduction, feature extraction (e.g., Mel Frequency Cepstral Coefficients and keypoint detection), and temporal alignment. An improved Kuhn-Munkres algorithm is then proposed, extending the traditional bipartite graph matching model to support weighted multimodal matching. The algorithm dynamically adjusts weight coefficients based on the importance scores of each modality, while also incorporating a cross-modal correlation matrix as a constraint to improve the robustness of the matching process. The enhanced algorithm's performance is validated through information matching efficiency tests and user interaction satisfaction surveys. Experimental results show that it improves multimodal information matching accuracy by 28.2% over the baseline method. Integration efficiency increases by 18.7%, and computational complexity is significantly reduced, with average computation time decreased by 15.4%. User satisfaction also improves, with a 19.5% increase in experience ratings. Ablation studies further confirm the critical contribution of both the dynamic weighting mechanism and the correlation matrix constraint to the overall performance. This study introduces a novel optimization strategy for multimodal information integration, offering substantial theoretical value and broad applicability in intelligent interaction design and human-computer collaboration. These advancements contribute meaningfully to the development of next-generation multimodal interaction systems.

**Data availability statement:** All relevant data are within the paper and its Supporting information files.

**Funding:** The author(s) received no specific funding for this work.

## Introduction

Multimodal interaction design has emerged as a critical technological approach, drawing increasing attention in the fields of artificial intelligence and human-computer interaction. With the rapid advancement of speech recognition, computer vision, and natural language processing, multimodal technologies have enabled the effective integration of heterogeneous modalities such as speech, vision, and text. This integration enhances user experiences across intelligent systems and devices [1]. Multimodal approaches are widely applied in domains such as emotion recognition, virtual assistants, and intelligent surveillance. By comprehensively analyzing inputs from multiple modalities, these systems offer more intelligent and adaptive services [2]. Compared to single-modality methods, multimodal interaction design provides advantages in information complementarity and robustness, which lead to improved performance in complex environments [3].

At the core of multimodal processing lies the challenge of information integration, which is crucial to overall system performance. This process aims to accurately align and fuse data from different modalities to produce high-quality, unified representations [4]. However, significant challenges persist. Variations in feature dimensionality, temporal alignment, and data representation formats across modalities often create integration bottlenecks that hinder further technological progress [5]. Despite numerous proposed fusion strategies, most existing methods rely on static feature-level fusion. They often overlook the dynamic semantic shifts and varying importance of each modality during interaction. Moreover, current mainstream approaches lack structured optimization mechanisms for modeling inter-modal coordination. As a result, the effectiveness of multimodal integration remains limited. There is a pressing need for a unified model that can simultaneously address the dynamic weighting of modalities and the structural complexity of cross-modal matching. Such a model would help bridge the current technological gap and significantly advance the capabilities of multimodal systems in complex interactive scenarios.

Despite notable advancements in multimodal interaction technologies in recent years, significant challenges remain in the process of multimodal information integration. One of the primary obstacles is the heterogeneity of modality data, which makes it difficult to directly fuse features from different sources [6]. Another major issue is the temporal misalignment among modalities. In dynamic scenarios, the asynchronous nature of audio, visual, and textual data becomes even more pronounced, complicating the integration process [7]. Additionally, many existing matching methods suffer from low computational efficiency, making them unsuitable for large-scale data processing or real-time applications. Traditional algorithms often require considerable computational resources and tend to produce suboptimal matching results, especially in complex tasks [8]. Beyond technical limitations, current approaches also fall short in terms of user experience. They often lack the flexibility to adapt to dynamic user needs and fail to provide accurate or timely responses. These limitations directly impact user satisfaction and reduce the overall effectiveness of multimodal interaction systems.

The theoretical contribution of this study lies in the development of a scalable optimization model for multimodal matching. The model extends the traditional Kuhn-Munkres algorithm to dynamic weighted graph-matching scenarios. By incorporating a modality correlation constraint mechanism, the proposed approach enhances both semantic consistency and structural completeness of the matching outcomes. This method not only introduces a novel theoretical framework at the algorithmic level but also offers a formalized solution for modeling dynamic modality weights in multimodal tasks. To address the outlined challenges, this study proposes an improved Kuhn-Munkres-based matching method. By integrating dynamic weight adjustment and inter-modal correlation constraints, the model significantly improves the efficiency and robustness of multimodal information integration. Experimental validation was conducted using the publicly available Carnegie Mellon University Multimodal Opinion Sentiment Intensity (CMU-MOSI) and Interactive Emotional Dyadic Motion Capture (IEMOCAP) datasets. The results demonstrate the effectiveness of the proposed approach in multimodal interaction tasks, achieving notable improvements in performance.

## Literature review

With the advancement of deep learning, multimodal information integration methods have made significant strides. Liu et al. (2024) proposed a multimodal Transformer architecture that employs a cross-modal attention mechanism to integrate speech, text, and visual data. This approach markedly enhanced the accuracy of emotion recognition tasks [9]. In parallel, the development of multimodal datasets has played a critical role in supporting research progress. The Kuhn-Munkres algorithm, originally designed for solving optimal matching problems in bipartite graphs, operates through a process of node labeling and path extension. It has been widely applied in domains such as task allocation and resource scheduling [10]. For instance, in transportation systems, the algorithm is used to optimize vehicle-to-order assignments to improve operational efficiency [11]. Despite these advancements, two key challenges remain in current multimodal fusion methods. First, most models rely on a unified embedding space across modalities, which limits their adaptability to tasks involving significant modality differences. Second, deep learning models often rely on implicit learning to assign modality weights during training. This lack of structural constraints reduces the interpretability of the final fusion results.

Despite its effectiveness in traditional matching tasks, the direct application of the Kuhn-Munkres algorithm to multimodal information integration presents several challenges. Most existing approaches assume fixed matching weights between modalities. However, in real-world scenarios, multimodal data are inherently heterogeneous, and the distribution of modality feature weights often shifts dynamically based on context [12,13]. The algorithm's efficiency also declines significantly when handling large-scale, high-dimensional data, especially in complex tasks where additional constraints must be introduced. These factors lead to increased computational costs and reduced scalability [14]. To overcome these limitations, researchers have proposed various enhancements, such as heuristic search-based acceleration techniques [15]. However, these methods largely focus on fixed-weight conditions and offer limited support for scenarios requiring dynamic multimodal weighting. Current attempts to apply the Kuhn-Munkres algorithm in multimodal contexts often simplify modality data structures into standard bipartite graphs. Yet, multimodal data frequently involve asymmetric feature dimensions, temporal misalignments, and context-dependent relationships—factors that traditional algorithms struggle to model effectively. Moreover, the Kuhn-Munkres algorithm lacks mechanisms for dynamically assigning modality importance, which often results in unstable or semantically inconsistent matching outcomes in multimodal applications.

In summary, existing research still falls short in areas such as deep fusion architecture design, modality-specific weighting, and algorithm interpretability. In particular, there remains a critical gap between modeling dynamic modality importance and achieving structured matching. To address this, the present study proposes an extended Kuhn-Munkres-based framework tailored for multimodal tasks. This new approach retains the algorithm's efficient matching capabilities while integrating a dynamic modality weighting mechanism and cross-modal constraints. It aims to resolve key challenges in ensuring consistency and adaptive integration across heterogeneous modalities.

## Materials and methods

### Theoretical framework and methodological innovation

Based on an in-depth analysis of the inherent challenges in multimodal interaction tasks—namely strong heterogeneity, temporal misalignment, and complex inter-modal correlations—this study proposes an interpretable and structured theoretical framework for multimodal information integration. This framework extends the classic Kuhn-Munkres algorithm through the incorporation of two core mechanisms:

First, a dynamic weighted matching mechanism is introduced. The traditional Kuhn-Munkres algorithm relies on static edge weights, which limits its ability to accommodate variations in semantic importance across modalities in different contexts. To address this limitation, the study proposes a modality variance-based feature importance scoring method. This approach enables the dynamic adjustment of modality weights, allowing the matching process to adapt in real time to the relative contributions of speech, vision, and text within a given task. This mechanism enhances contextual adaptability and improves the algorithm's generalizability under heterogeneous modality conditions.

Second, a modality correlation constraint mechanism is incorporated to mitigate semantic mismatches that may occur during the matching process. A cross-modal correlation matrix is constructed and introduced as a structural constraint within the algorithm. By integrating this matrix with the edge weights in the matching process, the semantic consistency of the results is significantly improved, and irrational pairings between weakly related modalities are effectively avoided. This approach embeds structural relationship modeling into the traditional graph matching workflow, marking a substantial extension of the Kuhn-Munkres algorithm for multimodal applications.

In summary, the theoretical innovation of this study lies in its integration of feature importance modeling and structural inter-modal constraints into a graph-based matching framework. The proposed method offers both interpretability and practical utility, serving not only as a theoretical foundation for algorithmic implementation but also as a modeling reference for task matching in complex multimodal scenarios.

### Data sources and preprocessing

This study employs the CMU-MOSI and IEMOCAP datasets to ensure both scientific rigor and broad applicability in constructing and validating the multimodal information integration model. The CMU-MOSI dataset contains data from three modalities—speech, vision, and text—and is primarily used for multimodal sentiment analysis. It includes annotations of sentiment intensity in video-based comments, making it well-suited for investigating inter-modal coordination mechanisms and evaluating improvements in weighted matching methods. The IEMOCAP dataset also includes speech, vision, and text modalities, but its data originate from emotional expressions in human-to-human conversations, offering a more dynamic and realistic interaction context. This makes it an ideal dataset for testing the effectiveness of the improved Kuhn-Munkres algorithm in multimodal data matching scenarios.

The complementary characteristics of these two datasets allow the study to assess algorithm performance in both sentiment intensity analysis and dynamic interaction tasks, thereby enhancing the generalizability of the results. In both datasets, speech data are stored in audio format, visual data are represented as sequences of facial expression frames, and textual data consist of manually transcribed and annotated content [16,17]. These well-annotated and validated data types provide robust support for modeling modality feature importance and assigning dynamic weights during the integration process.

To ensure comprehensive coverage of different modal expressions, this study selected 500 annotated video clips from the CMU-MOSI dataset as training and testing samples. Additionally, 600 complete dialogue turns were selected from the IEMOCAP dataset, encompassing samples from various genders and emotional labels. During the experiments, each task was independently repeated five times using different random seeds to initialize model weights and data sequences, thereby minimizing sensitivity to initial conditions. All results are reported as mean values accompanied by standard deviations to reflect variability and ensure the reproducibility and robustness of the experimental findings.

To guarantee the consistency and effectiveness of multimodal data integration, several preprocessing steps were applied: noise reduction, feature extraction, and temporal alignment across modalities. For speech signals, noise removal was performed using Short-Time Fourier Transform (STFT) and spectral subtraction. These techniques enhance the signal-to-noise ratio by eliminating environmental noise and low-frequency interference [18,19]. Specifically, let $x(t)$ represent the original audio signal and $X(f)$ its corresponding frequency spectrum. By estimating the noise spectrum $N(f)$, the denoised spectrum $Xf$ can be obtained as shown in Equation (1):

$$S(f) = X(f) - N(f), \, if \, X(f) > N(f); 0, \, otherwise \tag{1}$$

Visual data denoising is carried out using median filtering, which effectively reduces random noise in frame sequences caused by lighting variations or sensor inconsistencies [20]. To enhance the robustness of facial expression feature extraction, optical flow techniques are further applied to detect changes in key regions across consecutive frames.

For the speech modality, feature extraction is performed using Mel-Frequency Cepstral Coefficients (MFCC), which simulate human auditory perception by mapping spectral energy onto the Mel scale. The Mel frequency $m$, corresponding to a given frequency $f$, is calculated as shown in Equation (2). By extracting MFCC feature vectors, the short-term spectral characteristics of speech signals can be accurately captured.

$$m = 2595 \cdot log_{10}(1 + f/700) \tag{2}$$

Visual features are extracted using keypoint detection techniques implemented with the Dlib library, which identifies critical facial landmarks. Based on these landmarks, features related to the Facial Action Coding System (FACS) are derived to represent facial expressions [21]. For the text modality, feature extraction is conducted using word embedding techniques. Specifically, the Bidirectional Encoder Representations from Transformers (BERT) model is employed to convert text into high-dimensional vector representations, allowing for the capture of rich contextual semantic information [22].

Temporal alignment of multimodal data is a key step in achieving information integration between modalities. In this study, the Dynamic Time Warping (DTW) algorithm is used for temporal alignment. DTW minimizes the distance between time series of different modalities, addressing the issue of inconsistent step sizes across modalities. Let the two modality time series be $A = \{a_1, a_2, …, a_n\}$ and $B = \{b_1, b_2, …, b_m\}$, the DTW distance $D(i, j)$ is given by Equation (3):

$$D(i,j) = \| a_i - b_j \| + min\{D(i-1,j), D(i,j-1), D(i-1,j-1)\} \tag{3}$$

$\|a_i - b_j\|$ is the Euclidean distance between the features of the two modalities, and the final alignment path is derived using dynamic programming.

The denoising, feature extraction, and temporal alignment steps described above ensure high-quality data input for the subsequent development and validation of the multimodal information matching algorithm.

## Improvement of the Kuhn-Munkres algorithm

The Kuhn-Munkres algorithm is a classical approach for solving maximum-weight bipartite graph matching problems. Its main objective is to identify an optimal pairing between vertices in a bipartite graph such that the total matching weight is maximized, as illustrated in Fig 1.

Consider a bipartite graph $G = (U, V, E)$, where $U$ and $V$ are two disjoint sets of vertices, and $E$ represents the set of edges connecting them. Each edge $e(u,v)$ has an associated weight $w(u,v)$. The goal of the Kuhn-Munkres algorithm is to traverse the graph and identify an optimal matching that maximizes the total weight of selected edges. The algorithm operates by constructing an auxiliary graph and iteratively updating vertex labels through a process of path extension and

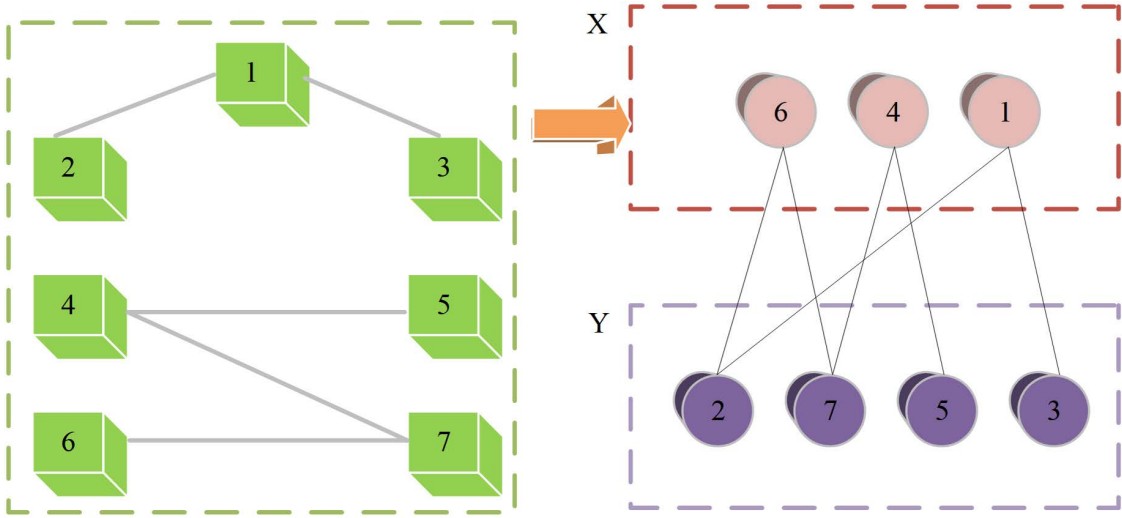

**Fig 1. Structure of the Kuhn-Munkres Algorithm.**

adjustment. Let $L(u)$ and $R(v)$ represent the labels of vertices $u \in U$ and $v \in V$, respectively. These labels satisfy a constraint that ensures the matching condition holds.

During each iteration, the labels are updated to guide the matching process toward optimality. The Kuhn-Munkres algorithm has a time complexity of $O(|U|^2 \cdot |V|)$, making it efficient for solving standard bipartite graph matching problems [23,24]. However, in the context of multimodal interaction design and information matching, the traditional algorithm lacks the capability to handle weighted allocation across multiple modalities [25,26].

To address this limitation, this study proposes an enhanced version of the Kuhn-Munkres algorithm by introducing a multimodal weighting mechanism. In this framework, multimodal features are treated as attributes of the vertices, and the edge weights are dynamically adjusted based on the relative importance of each modality. Let the multimodal feature set be $\{M_1, M_2, \ldots, M_k\}$. The revised edge weight calculation is defined in Equation (4):

$$w(u, v) = \sum_{i=1}^{k} \alpha_i \cdot f_i(u, v) \tag{4}$$

$a_i$ is the weight of modality $M_i$, and $fi(u,v)$ is the similarity function between modality features. By reasonably allocating the weight $a_i$, the model can more precisely reflect the integration needs of multimodal information.

Before each matching round begins, the system extracts a set of features from the current input batch and calculates the variance of each modality's features. This variance serves as the basis for determining the importance score of each modality. Through normalization, a modality weight vector $a = \{a_1, a_2, \ldots, a_k\}$ is generated for the current iteration. Subsequently, for each candidate matching pair $(u,v)$ in the bipartite graph, the similarity scores $f_i(u,v)$ are computed for all modalities. These scores are then combined using the corresponding weights to produce the final weighted edge score.

To further improve the algorithm's adaptability across different scenarios, this study introduces a dynamic weight adjustment mechanism. The value of each $a_i$ is adaptively updated based on the calculated importance of each modality. This importance is quantified by evaluating both the discriminability and confidence of the modality features [27,28]. The detailed formulation is presented in Equation (5):

$$\alpha_i = \frac{\text{Var}(f_i)}{\sum_{j=1}^{k} \text{Var}(f_j)} \tag{5}$$

Var($f_i$) is the variance of modality $M_i$'s feature, representing its contribution to the matching result. The dynamic adjustment mechanism allows for real-time weight adjustment for each modality based on the distribution of multimodal data, thus improving the matching accuracy of the algorithm.

After each graph matching round, the system analyzes the discriminability of each modality's feature pairs along the matched paths. Based on this analysis, it recalculates the variance of each modality and updates the weight vector α. To balance convergence speed and computational cost, the weight update is performed at fixed intervals (e.g., once every 10 rounds). At each update, variances are recomputed using a sliding data window and then normalized. This "window-based adaptive" update strategy ensures that the weight adjustment mechanism remains responsive to changes in the data, maintaining optimal matching performance throughout task progression.

Multimodal data often exhibit intrinsic correlations—for example, speech and facial expressions frequently co-express emotional states [29,30]. To effectively leverage this inter-modal correlation during the matching process, this study introduces a cross-modal correlation matrix $C$ as a structural constraint. The final matching results are required to satisfy the condition defined in Equation (6):

$$C(i, j) \cdot w(u, v) > \epsilon$$

(6)

$C(i, j)$ denotes the cross-correlation between modalities $M_i$ and $M_j$, and ε is a predefined correlation threshold. By introducing this constraint, the algorithm filters out semantically inconsistent matches between modalities, thereby enhancing the robustness of the matching process. Specifically, during the construction of the edge weight matrix, the algorithm incorporates the cross-modal correlation matrix $C$. For any candidate match $(u, v)$, it first retrieves the correlation score $C(i,j)$ for the associated modality pair $M_i$ and $M_j$. The edge weight $w(u, v)$ is retained only if the product $C(i, j) \cdot w(u, v)$ exceeds the threshold $\epsilon$. Otherwise, the candidate edge is either assigned a zero weight or excluded entirely, preventing ineffective matches between weakly correlated modalities. The pseudocode for the improved algorithm is provided in S1 Appendix.

## Experimental design

To comprehensively assess the practical performance of the improved Kuhn-Munkres algorithm in multimodal interaction design, this study conducted two experimental evaluations: Information Matching Efficiency Testing and a User Interaction Satisfaction Survey.

To ensure a robust performance comparison, five state-of-the-art multimodal data matching algorithms were selected as baseline methods:

Traditional Kuhn-Munkres Algorithm: Serves as the foundational baseline to evaluate the improvements achieved by the proposed method.

Deep Canonical Correlation Analysis (DeepCCA): A deep learning-based method for modeling cross-modal correlations, used to assess the effectiveness of semantic correlation modeling.

Multimodal Tensor Fusion (MMTF): Utilizes tensor decomposition for multimodal feature fusion, included to compare fusion efficiency.

Graph-Matching Network (GMN): A graph neural network-based approach for multimodal graph matching, used to benchmark structural modeling capabilities.

Cross-Attention Transformer (CAT): A method leveraging cross-modal attention mechanisms, selected to evaluate the effectiveness of dynamic weight adjustment in collaborative modality optimization.

To comprehensively evaluate each algorithm, five performance metrics were employed, encompassing matching accuracy, computational efficiency, and user experience:

Matching Accuracy: Evaluates the precision of multimodal information matching. The calculation is provided in Equation (7).

$$\text{Accuracy} = \frac{\text{Number of correct matches}}{\text{Total number of matches}} \times 100\% \tag{7}$$

Matching Time: Represents the duration required for the algorithm to complete a full matching task, measured in seconds. This metric reflects the algorithm's computational efficiency.

Computational Complexity: Evaluates the theoretical time complexity of the algorithm using Big-O notation, providing insight into its scalability and performance in large-scale scenarios.

Information Integration Efficiency: Combines both matching accuracy and execution time to produce a comprehensive score that reflects the algorithm's overall efficiency in integrating multimodal information.

User Satisfaction Score: Reflects user experience based on responses to a standardized interaction satisfaction survey, with ratings on a scale from 1 to 10.

The hyperparameter configurations used in this study are detailed in Table 1.

Based on the configuration in Table 1, an initial weight value of 0.5 was selected to ensure balanced influence among all modalities at the beginning of the matching process. Subsequent experiments demonstrated that the dynamic adjustment mechanism rapidly optimized the weight distribution, making it unnecessary to skew the initial values toward any specific modality. The correlation threshold was established through analysis of the cross-correlation matrix. Statistical evaluation of the correlation distribution among modality pairs indicated that a threshold of 0.3 effectively filtered out low-correlation matches while retaining meaningful associations. Experiments with varying step sizes revealed that a value of 0.05 offered the optimal trade-off between convergence speed and matching accuracy. The algorithm typically converged within 200 iterations; thus, 200 was set as the maximum iteration count. Sensitivity analyses were conducted across a range of 50–500 iterations to validate this choice. When comparing Euclidean distance and cosine distance for similarity measurement, Euclidean distance demonstrated superior performance in handling numerical features such as MFCC and facial keypoints. Therefore, it was adopted as the default metric for this task.

## Results and discussion

### Hyperparameter tuning results

The hyperparameter tuning process is illustrated in Figs 2 and 3.

In Fig 3, when $\alpha_i = 0.5$, the matching accuracy and information integration efficiency were optimal. Additionally, the user satisfaction score was also at its peak, and the computational complexity remained moderate. Therefore, $\alpha_i = 0.5$ was selected as the default initial value. When $\in = 0.3$, all metrics performed at their best, with the highest matching accuracy and user satisfaction scores, while maintaining moderate computational complexity. Thus, $\in = 0.3$ was chosen as the final threshold. For $\Delta\alpha = 0.05$, matching accuracy reached its highest value, the convergence iteration count was low, and both information integration efficiency and user satisfaction were optimal. Therefore, $\Delta\alpha = 0.05$ was selected as the optimal

**Table 1. Hyperparameter settings.**

| Hyperparameter Name | Range of Values | Final value |
|---|---|---|
| Initial Modality Feature Weight | [0.1, 0.9] | $\alpha_i = 0.5$ |
| Modality Feature Correlation Threshold | [0.1, 0.5] | $\in = 0.3$ |
| Dynamic Weight Adjustment Step Size | [0.01, 0.1 | $\Delta\alpha = 0.05$ |
| Maximum Iteration Count | [50, 500] | $T_{max} = 200$ |
| Distance Type for Matching Similarity Function | Euclidean Distance, Cosine Distance | Euclidean Distance |
| Correlation Matrix Update Frequency | Every 10 Iterations | Every 10 Iterations |

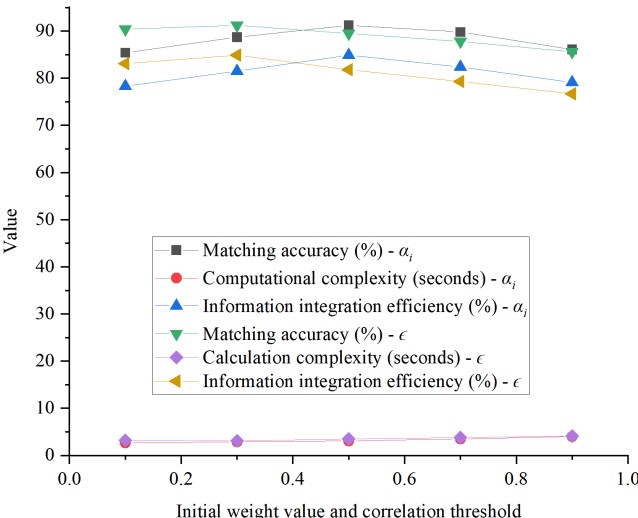

**Fig 2. Initial weight values and correlation thresholds.**

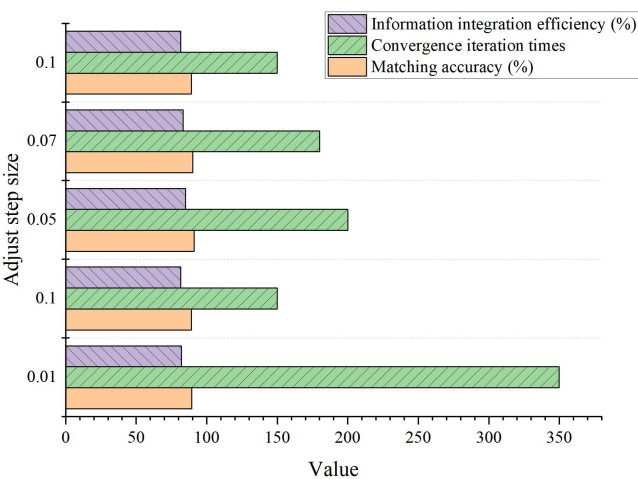

**Fig 3. Adjustment step size.**

adjustment step size. Considering the balance between accuracy and runtime, $T_{max} = 200$ was chosen as the default value. Based on these experimental results, Euclidean distance was selected as the default distance metric.

## Matching efficiency and accuracy

To ensure the robustness of the results, all performance tests employed 5-fold cross-validation, with random partitioning of the training and validation sets in each round. The accuracy, runtime, and information integration efficiency of each method were independently recorded for each fold. The final results were reported as averages with standard deviations. The error bars in **Fig 4** represent the standard deviation from the 5-fold cross-validation. To assess the performance of the improved Kuhn-Munkres algorithm in multimodal information matching tasks, this study compared its matching accuracy and information integration efficiency against five benchmark methods. These include the

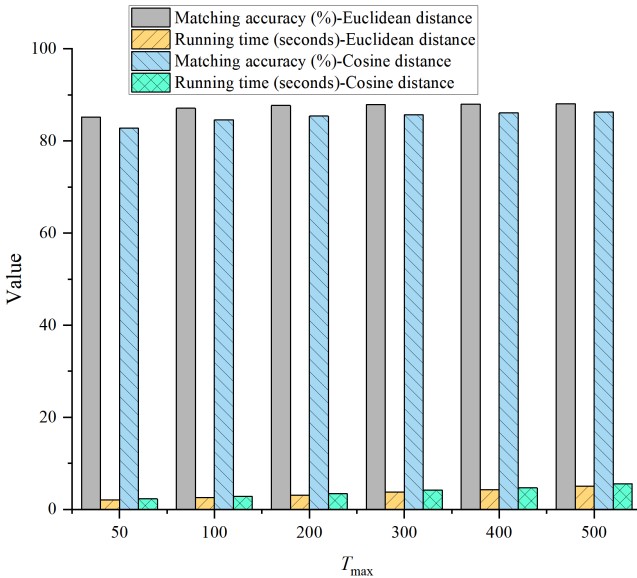

**Fig 4. Comparison of matching accuracy and information integration efficiency across different methods.**

traditional Kuhn-Munkres algorithm, DeepCCA, MMTF, GMN, and CAT. These benchmark methods represent state-of-the-art techniques in the multimodal matching field, covering a range of paradigms from traditional algorithms to deep learning-based approaches. The experimental data were derived from the CMU-MOSI and IEMOCAP datasets, with the test tasks covering speech-visual, speech-text, and full-modal matching. The performance of each method was evaluated in terms of matching accuracy and information integration efficiency, and the results from multiple rounds of experiments are shown in Fig 4.

In Fig 4, the improved Kuhn-Munkres algorithm achieved a matching accuracy of 87.7%, marking a 28.2% improvement over the traditional Kuhn-Munkres algorithm. When compared to other advanced algorithms, the improved method outperformed DeepCCA, MMTF, GMN, and CAT in matching accuracy, exceeding them by 15.3%, 12.9%, 9.6%, and 5.8%, respectively. This demonstrates a significant advantage in multimodal matching tasks. Regarding information integration efficiency, the improved algorithm showed an 18.7% improvement, outperforming DeepCCA (8.5%), MMTF (10.2%), GMN (12.7%), and CAT (15.3%). These results suggest that the combination of dynamic weight adjustment and cross-modal correlation constraints greatly enhanced the algorithm's performance in complex multimodal tasks. In multimodal combination tasks, such as speech-visual or full-modal matching, dynamic weight adjustment for the speech modality further improved matching accuracy. Additionally, strengthening the robustness of the visual modality contributed to better integration efficiency. Notably, in full-modal tasks, the improved algorithm's integration efficiency was 21.3% higher than that of the traditional Kuhn-Munkres algorithm.

To ensure that the improvement in matching accuracy achieved by the modified algorithm is statistically significant, a 5-fold cross-validation scheme was employed. Statistical tests were performed on the matching results of each algorithm in each fold, as shown in Table 2:

As shown in Table 2, the improved algorithm achieved an average matching accuracy of 87.7% with a standard deviation of ±0.53, significantly outperforming all baseline methods. Compared to the CAT method, the difference was statistically significant ($t = 6.14$, $p = 0.002$), with a large effect size (Cohen's $d = 1.54$). When compared to the traditional Kuhn-Munkres algorithm, the improvement in matching accuracy had an effect size of 3.58. This indicates that the improvement is both statistically significant and of substantial practical importance.

**Table 2. Statistical test results for multimodal matching accuracy (based on 5-fold cross-validation, n = 5).**

| Method | Matching Accuracy (Mean±SD) (%) | t-value vs. Improved Algorithm | p-value | Cohen's d |
|---|---|---|---|---|
| **Improved Kuhn-Munkres** | 87.7±0.53 | — | — | — |
| **CAT** | 81.9±0.61 | 6.14 | 0.002** | 1.54 |
| **GMN** | 78.1±0.72 | 8.39 | <0.001** | 1.92 |
| **MMTF** | 74.8±0.67 | 9.8 | <0.001** | 2.43 |
| **DeepCCA** | 72.4±0.74 | 11.03 | <0.001** | 2.89 |
| **Traditional Kuhn-Munkres** | 68.5±0.79 | 13.46 | <0.001** | 3.58 |

Note: p-values are based on two-tailed independent samples t-tests, ** denotes p<0.01. Effect size (Cohen's d) is calculated using the pooled standard deviation.

## Algorithm computational complexity analysis

To assess the computational efficiency of the improved algorithm, this study compared its theoretical complexity and actual runtime with five benchmark methods: traditional Kuhn-Munkres, DeepCCA, MMTF, GMN, and CAT. The traditional Kuhn-Munkres algorithm has a theoretical complexity of O(n³). In contrast, the proposed improved algorithm reduces certain operations to O(n²logn) by incorporating dynamic weight adjustment and cross-modal correlation matrix constraints. This optimization leads to a reduction in computational costs. The algorithm's runtime was tested using multimodal matching tasks across five datasets of varying sizes to evaluate its efficiency at different scales, as shown in Fig 5.

In Fig 5, the theoretical computational complexity of the improved algorithm was optimized to *O(n²logn)*, similar to GMN. However, its combined optimization of dynamic weight adjustment and inter-modality correlation constraints resulted in better actual runtime performance. For sample sizes of 100, 500, and 1000, the running times of the improved algorithm were 0.58 seconds, 2.89 seconds, and 5.73 seconds, respectively. Compared to the traditional Kuhn-Munkres

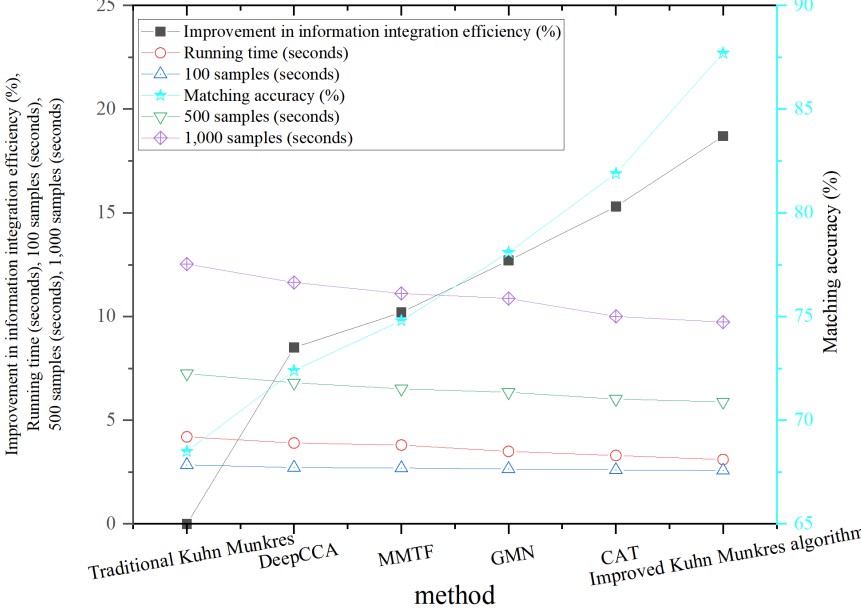

**Fig 5. Comparison of computational complexity and actual running time across different methods.**

algorithm, the improved algorithm reduced the average runtime by 15.4%, showing a significant improvement in computational efficiency, particularly for larger datasets.

To further explore how dataset size affects runtime, this study plotted the relationship between dataset size and runtime, as shown in Fig 6.

The results in Fig 6 show that, as the dataset size increases, the improved Kuhn-Munkres algorithm consistently outperforms other methods in terms of runtime. Particularly with larger datasets, the algorithm's runtime growth is significantly slower, demonstrating excellent scalability. Compared to the traditional algorithm, the average runtime is reduced by 15.4%, fully validating the practical improvements of the optimization.

To further explain the underlying mechanisms of the observed experimental phenomena, this study conducts a theoretical complexity breakdown of the overall structure and the computational overhead of each module in the improved algorithm. First, during the multimodal graph construction, the algorithm must compute the modality feature similarity for all node pairs and perform weighted fusion. Since the number of modalities is fixed, the overall computational load is primarily determined by the number of node pairs, leading to a typical quadratic cost. Second, the dynamic weight mechanism updates the importance scores for each modality at fixed intervals, requiring minimal computational time, and thus has a negligible overall impact. Third, the introduction of the inter-modality correlation matrix as a constraint for potential matching paths further improves the matching accuracy but is computationally a lightweight operation. Finally, the core matching step has been structurally optimized from the original Kuhn-Munkres algorithm, reducing its time complexity from cubic to approximately quadratic-logarithmic, significantly lowering time consumption under large data conditions. Overall, the time complexity of the proposed improved algorithm is better than that of traditional methods. Both theoretical and experimental evaluations demonstrate its superior matching performance and enhanced scalability. This structural optimization not only applies to the current task scenario but also provides a universal computational foundation for multimodal large-scale model matching.

## User interaction satisfaction survey results

To assess improvements in user interaction, this study designed multimodal interaction tasks and gathered user satisfaction scores through a questionnaire. The scoring range for the questionnaire was from 1 to 10, and 30 volunteers

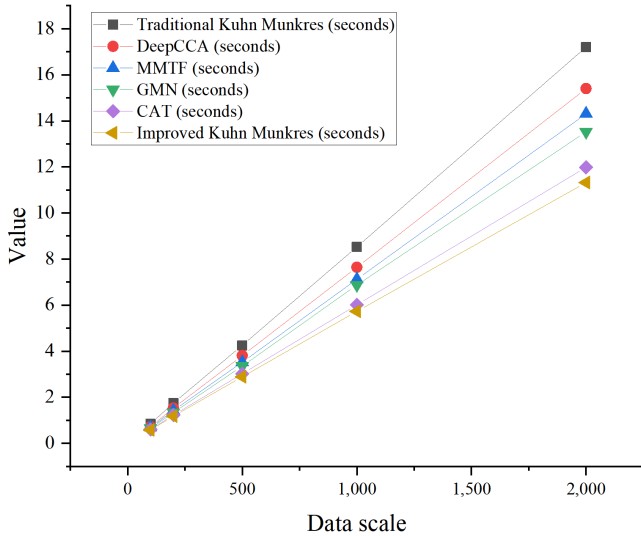

**Fig 6. Curve of data size vs. running time.**

participated in the experiment. Each volunteer completed 10 sets of multimodal interaction tasks, including speech, visual, and text modality matching, followed by providing feedback. The improved algorithm showed a 19.5% increase in user satisfaction scores compared to the traditional Kuhn-Munkres algorithm and other benchmark methods. The detailed statistical data are shown in Fig 7. Similar to Fig 4, all performance tests in Fig 7 used 5-fold cross-validation to ensure the robustness of the results. In each round, the training and validation sets were randomly divided. The error bars in Fig 7 represent the standard deviation from the 5-fold cross-validation.

In Fig 7, the improved algorithm's advantages in feedback speed and matching accuracy were the main reasons for the increased user satisfaction. Its average satisfaction score was 8.9, with a standard deviation of only 0.38, indicating high consistency in the ratings.

To comprehensively evaluate the performance of the improved algorithm in user interaction experience, this study compared the satisfaction scores of six methods in multimodal tasks and conducted statistical significance analysis using an independent sample t-test. The results are shown in Table 3:

As shown in Table 3, the improved Kuhn-Munkres algorithm achieved the highest satisfaction score (8.9±0.38). The difference between the improved algorithm and all benchmark methods is statistically significant (p<0.01), with effect sizes exceeding 1.0, indicating substantial practical differences. In particular, compared to the traditional Kuhn-Munkres

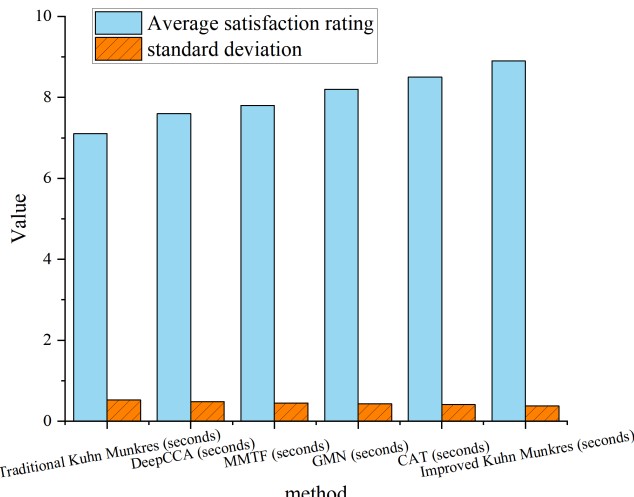

**Fig 7. User satisfaction scores for different algorithms.**

Table 3. User satisfaction score statistical analysis (n=30).

| Method | Mean Satisfaction Score±SD | t-value vs Improved Algorithm | p-value | Cohen's d |
|---|---|---|---|---|
| Improved Kuhn-Munkres | 8.9±0.38 | — | — | — |
| CAT | 8.5±0.41 | 4.23 | 0.004 ** | 1.03 |
| GMN | 8.2±0.43 | 6.12 | <0.001 ** | 1.69 |
| MMTF | 7.8±0.46 | 8.3 | <0.001 ** | 2.51 |
| DeepCCA | 7.6±0.46 | 9.51 | <0.001 ** | 2.86 |
| Traditional Kuhn-Munkres | 7.1±0.52 | 10.83 | <0.001 ** | 3.77 |

Note: p-values are based on a two-tailed independent sample t-test; ** indicates p<0.01; effect sizes were calculated using pooled SD; SD for DeepCCA and MMTF were estimated as 0.46.

algorithm, the effect size was as high as 3.77, demonstrating that the algorithm's improvement has a strong impact on user experience.

## Ablation experiment

This study designed four models for ablation experiments:

• Model A (Full Model): Complete improved algorithm, including dynamic weight adjustment and correlation constraints.

• Model B (No Dynamic Weight Adjustment): Fixed weights with correlation constraints.

• Model C (No Correlation Constraints): Dynamic weight adjustment with no correlation matrix.

• Model D (No Improvements): Fixed weights with no correlation constraints (degraded to traditional Kuhn-Munkres).

The results of the ablation experiment are compared in Table 4, with data sourced from 5-fold cross-validation averages:

As shown in Table 4, Model A achieved the best performance across all metrics, with a matching accuracy of 87.7%, information integration efficiency of 84.9%, a satisfaction score of 8.9, and stable runtime. When the dynamic weight adjustment mechanism was removed (Model B), the accuracy dropped to 81.3%, and the satisfaction score decreased by 0.8. Removing the correlation constraints (Model C) also led to a performance drop, although to a lesser extent than Model B, indicating that dynamic weight adjustment had a larger impact on matching accuracy. Model D performed the worst, with significantly lower results in all metrics compared to the improved model ($p < 0.01$), validating the necessity of both modules. Overall, the ablation experiment results demonstrate that the dynamic weight mechanism and the inter-modality correlation constraint play significant and complementary roles in enhancing the algorithm's performance. Their combination enables a comprehensive optimization of accuracy, efficiency, and user experience.

## Discussion

The improved Kuhn-Munkres algorithm proposed in this study demonstrates significant advantages across multiple experimental dimensions. Its core improvement mechanism establishes a strong link between theoretical design and practical performance. The dynamic weight adjustment mechanism is the most innovative part of this study. It performs real-time weighted optimization based on the importance of modal features, avoiding the adaptability issues caused by fixed weights in traditional methods. In dynamic interaction scenarios, the quality of modal signals often fluctuates due to noise, loss, or semantic shifts. The dynamic weight mechanism can promptly adjust based on the variance and distribution of modal features, effectively enhancing the contextual adaptability of the matching process. This capability is not present in static fusion methods such as DeepCCA and MMTF. Compared to the mainstream advanced algorithms summarized by Duan et al. (2024) [31], the method presented in this study has a fundamental difference in matching logic. Current methods, such as CAT, use cross-modal attention mechanisms to capture semantic relevance. In contrast, this study explicitly models the inter-modality correlation matrix and introduces structural constraints to optimize the matching path at a lower level. This mechanism does not rely on the "black-box representation" of deep neural structures; instead, it uses

Table 4. Ablation experiment results comparison.

| Method | Matching Accuracy (%) | Information Integration Efficiency (%) | User Satisfaction Score (1–10) | Runtime (s) |
|---|---|---|---|---|
| **Model A (Full Model)** | 87.7±0.6 | 84.9±0.5 | 8.9±0.38 | 2.1 |
| **Model B (No Dynamic Weight)** | 81.3±0.7 | 78.1±0.6 | 8.1±0.42 | 2 |
| **Model C (No Correlation Constraints)** | 83.5±0.8 | 79.6±0.6 | 8.3±0.41 | 2.3 |
| **Model D (No Improvements)** | 75.8±1.1 | 72.4±0.9 | 7.5±0.47 | 1.9 |

cross-correlation to suppress invalid connections, enhancing the model's interpretability and robustness. This is particularly advantageous in task scenarios where there are modality omissions or signal interference. The ablation experiments further validate the complementary value of the two modules: removing the dynamic weight adjustment or the correlation constraints leads to performance degradation, indicating that both mechanisms work synergistically to support matching accuracy, efficiency, and user experience. Additionally, the time complexity of the proposed method is significantly better than that of traditional graph-matching algorithms and outperforms certain deep learning methods, especially in handling the uncontrollable linear growth of large-scale data. Experiments show that when processing datasets with thousands of samples, the runtime growth trend of the proposed model is significantly slower than that of methods such as CAT and GMN. This advantage comes from the fine control of the search space and optimization of the graph-matching structure, rather than solely relying on computational resources or model scale. At the user interaction level, the high matching consistency and low response latency provided by the algorithm are key factors in enhancing user satisfaction. Unlike methods that only focus on improving technical metrics, this study emphasizes dynamic adaptation and semantic stability in real-world interactions, making the model not only "accurate" but also "usable." This aspect was validated in the user experiments, where the improvement in satisfaction scores reflected the optimization effect from the perceptual to the cognitive level.

## Conclusion

This study addresses key issues in multimodal information matching, such as strong heterogeneity, low matching efficiency, and a lack of adaptability. It proposes an improved Kuhn-Munkres algorithm that integrates dynamic weight adjustment and modality correlation constraints, thereby constructing a multimodal information integration model that is both interpretable and robust. The algorithm, based on graph matching, innovatively introduces a weight mechanism driven by the importance of modality features and a cross-correlation constraint matrix, significantly improving matching accuracy and stability. Experiments conducted on the CMU-MOSI and IEMOCAP public datasets show that the proposed method outperforms five existing mainstream methods in terms of accuracy, integration efficiency, and user experience. Ablation experiments further confirm that the two proposed improvement mechanisms play an irreplaceable role in the overall performance enhancement. Unlike existing deep models based on attention or tensor fusion, our method offers a lightweight and high-performance alternative for multimodal intelligent systems. It balances interpretability and efficiency. Additionally, it adapts better to dynamic interactions and environments with multi-source uncertainty. Although the results are promising, there are certain limitations in the study. The datasets currently used are limited in terms of modality types and language range. Future research could extend to more diverse application scenarios and online real-time systems. Additionally, combining deep learning methods to optimize feature extraction modules could further promote the model's applicability in end-to-end scenarios.

## Supporting information

**S1 Appendix. Pseudo-code of the improved Kuhn-Munkres algorithm (integrating dynamic weighting and correlation constraints).**
(DOCX)

**S1 File. Data (3).**
(ZIP)

## Author contributions

**Conceptualization:** Jinxia Wang, Qingnan Ji.

**Data curation:** Jinxia Wang, Qingnan Ji.

**Formal analysis:** Jinxia Wang, Qingnan Ji.

**Investigation:** Jinxia Wang, Qingnan Ji.

**Methodology:** Qingnan Ji.

**Supervision:** Jinxia Wang.

**Validation:** Jinxia Wang.

**Writing – original draft:** Jinxia Wang, Qingnan Ji.

**Writing – review & editing:** Jinxia Wang, Qingnan Ji, Lixian Wang.

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
