## [Decision Letter · Decision Letter 0]

24 Mar 2025

PONE-D-25-06747
Information Integration Model in Interaction Design under Computer Multimodal Collaboration: The Integrated Application of Kuhn-Munkres Algorithm
PLOS ONE

Dear Dr. Wang,

Thank you for submitting your manuscript to PLOS ONE. After careful consideration, we feel that it has merit but does not fully meet PLOS ONE’s publication criteria as it currently stands. Therefore, we invite you to submit a revised version of the manuscript that addresses the points raised during the review process.

The reviewers have identified several critical areas requiring improvement:

Technical details: Please provide sample size clarification, replication details, and more thorough explanation of your algorithm's mechanisms.Statistical analysis: Include appropriate statistical tests with p-values, standard deviations, confidence intervals, and error bars in figures.Theoretical framework: More explicitly state your theoretical contribution, include an ablation study, and better highlight the research gap.Presentation quality: Improve figure resolution and clarity, provide formal algorithm complexity analysis, and enhance language quality throughout.

We look forward to receiving your revised manuscript.

Kind regards,

Yosi Kristian

Academic Editor

PLOS ONE

Additional Editor Comments:

The consensus from the three reviewers indicates several significant issues that need addressing:

Technical soundness issues:

Sample size and replication details are inadequate

The specific mechanisms of the improved Kuhn-Munkres algorithm aren't sufficiently detailed

The paper lacks crucial explanations to support its findings

Statistical rigor concerns:

Need for appropriate statistical tests with p-values

Missing standard deviations or confidence intervals

Lack of error bars in figures

Incomplete cross-validation procedure explanation

Novelty and contribution issues:

Theoretical framework needs clearer articulation

An ablation study is needed to demonstrate the contribution of each component

The research gap should be better highlighted in introduction and related work

Presentation quality problems:

Figure quality is poor (blurred and unclear images)

Language needs improvement (paragraph structure, clarity)

Formal algorithm complexity analysis is missing

Reviewers' comments:

Reviewer's Responses to Questions

**Comments to the Author**

1. Is the manuscript technically sound, and do the data support the conclusions?

Reviewer #1: Yes

Reviewer #2: Partly

Reviewer #3: No

2. Has the statistical analysis been performed appropriately and rigorously? 

Reviewer #1: Yes

Reviewer #2: Yes

Reviewer #3: No

3. Have the authors made all data underlying the findings in their manuscript fully available?

Reviewer #1: Yes

Reviewer #2: Yes

Reviewer #3: Yes

4. Is the manuscript presented in an intelligible fashion and written in standard English?

Reviewer #1: Yes

Reviewer #2: Yes

Reviewer #3: No

5. Review Comments to the Author

Reviewer #1: Some parts need to be organized and recycled in several paragraphs. Language needs more attention, in one paragraph there are at least 3 sentences. please add a few more sentences. The improved Kuhn-Munkres algorithm is described in general terms (such as the use of dynamic weights and correlation matrices), but the specific mechanism of how this improvement works is not described in sufficient detail.

Reviewer #2: This manuscript presents significant improvements to the Kuhn-Munkres algorithm for multimodal information integration in human-computer interaction. Below is a summary of strengths and required improvements to meet PLOS ONE standards:

Strengths

Clear methodological approach using established datasets (CMU-MOSI and IEMOCAP)

Innovative modifications to the Kuhn-Munkres algorithm (dynamic weight adjustment and cross-modal correlation constraints)

Comparative analysis against five baseline methods with quantitative results

Promising performance improvements (28.2% in matching accuracy, 18.7% in efficiency, 15.4% in computation time)

User experience validation (19.5% improvement in satisfaction ratings)

Required Improvements for Publication

Technical Soundness

Sample size clarification: Specify the number of test cases used from each dataset

Replication details: Indicate whether experiments were replicated and how variability was managed

Statistical Rigor

Statistical significance testing: Include appropriate statistical tests (t-tests or ANOVA) with p-values

Variability measurements: Add standard deviations or confidence intervals for all performance metrics

Error bars: Include these in figures to visualize variability

Cross-validation procedure: Explain the validation methodology (e.g., k-fold) and report performance across folds

Effect size analysis: Calculate and report effect sizes to contextualize improvements

Novelty Enhancement

Theoretical contribution: More explicitly state the theoretical framework advanced by this research

Ablation study: Demonstrate the relative contribution of each proposed component

Comparative discussion: Expand on fundamental differences from existing state-of-the-art methods

Visual and Presentation Quality

Figure quality: Improve resolution and clarity of all figures

Algorithm complexity analysis: Provide more formal and rigorous complexity analysis

Conclusion

This manuscript shows promise for publication in PLOS ONE with the implementation of the suggested improvements. The novel approach to multimodal information integration demonstrates practical significance, but requires enhanced statistical rigor and clearer articulation of theoretical contributions to fully meet publication standards.

Reviewer #3: The manuscript discusses the challenge of integrating heterogeneous data from speech, vision, and text in multimodal HCI systems. The aim is to improve accuracy, efficiency, and user experience in multimodal information integration tasks. The idea is interesting. However, the manuscript is not sufficient enough to be published in this journal as it lacks a lot of crucial explanations to support the findings.

The authors should also include theoretical analysis to better convey methodological improvements, include ablation studies, highlight the research gap in the introduction and related works, and provide more detailed reasoning to support the contribution with deep discussions.

The title also needs to be rewritten to better convey the proposed framework and the images are blurred and unclear.

6. PLOS authors have the option to publish the peer review history of their article (what does this mean?). If published, this will include your full peer review and any attached files.

Reviewer #1: **Yes: **Syafri Arlis

Reviewer #2: **Yes: **Adi Suryaputra Paramita

Reviewer #3: No

---

## [Author Response · Author response to Decision Letter 1]

17 Apr 2025

The reviewers have identified several critical areas requiring improvement:

1. Technical details: Please provide sample size clarification, replication details, and more thorough explanation of your algorithm's mechanisms.

Reply: Your suggestion is very good. We have added more explanations about the research details in the third section. Before each round of matching task starts, the system calculates the variance of each modality feature in turn (as the basis for importance scoring) based on the feature set extracted from the current input sample batch, and generates the modality weight vector α = {α₁, α₂, ..., αk} of the current round through normalization. Then, for each pair of candidate matches (u,v) in the bipartite graph, the similarity fi(u,v) under each modality is calculated respectively, and finally the weighted edge weight is obtained. We have added explanations about sample size and replication details in the second section of the method section. To ensure that the data fully covers different modal performances, this paper selects 500 annotated video clips from the CMU-MOSI dataset as training and test samples; 600 complete dialogue rounds are selected from IEMOCAP, including samples with different gender and emotion labels. During the experiment, all tasks are set up with 5 independent repeated experiments, and the weights and order are initialized with different random seeds to control the sensitivity of the model to the initial state. All results are reported as mean values with standard deviations to describe variability and ensure the replicability and robustness of the experiments.

2. Statistical analysis: Include appropriate statistical tests with p-values, standard deviations, confidence intervals, and error bars in figures.

Reply: Your comments are very objective. We have added these statistical analyses to the results section and modified the images in the results section. The average accuracy of the improved algorithm is 87.7%, with a standard deviation of ±0.53, which is significantly higher than all the baseline methods. Compared with the CAT method, the difference is statistically significant (t = 6.14, p = 0.002), and the effect size Cohen’s d = 1.54 is a large effect level. Compared with the traditional Kuhn-Munkres algorithm, the improvement in matching accuracy is as high as 3.58, indicating that the improved algorithm is not only statistically significant, but also has a substantial difference.

3. Theoretical framework: More explicitly state your theoretical contribution, include an ablation study, and better highlight the research gap.

Reply: Your suggestion is very good. We added a supplement about the theoretical framework in the first section of the method section. The theoretical innovation of this paper is reflected in: based on graph matching, it integrates feature importance modeling and inter-modal structural constraints. A multimodal information matching framework with both interpretability and practicality is proposed. This framework not only provides theoretical support for subsequent algorithm implementation, but also provides a modeling reference for subsequent task matching in multimodal scenarios.

We added ablation experiments in the results section. This paper designs the following 4 models for ablation experiments: A: full model (complete improved algorithm), specifically including dynamic weights + correlation constraints (this paper's model); B: remove dynamic weights, including fixed weights + correlation constraints; C: remove correlation constraints, including dynamic weights + no correlation matrix; D: remove all improvements, including fixed weights + no correlation (degenerated to traditional KM). Model A achieved the best performance in all indicators, with a matching accuracy of 87.7%, an information integration efficiency of 84.9%, a satisfaction score of 8.9, and a stable running time; after removing the dynamic weight mechanism (model B), the accuracy dropped to 81.3%, and the satisfaction score dropped by 0.8 points; removing the correlation constraint (model C) also showed a performance decline, but the degree was slightly less than that of B, indicating that the dynamic weight had a greater impact on the matching accuracy; Model D had the worst performance, and all indicators were significantly lower than the improved model (p<0.01), verifying the necessity of the two modules. In summary, the ablation experiment results show that the dynamic weight mechanism and the inter-modal correlation constraint have a significant and complementary role in improving the performance of the algorithm. The combination of the two can achieve comprehensive optimization of accuracy, efficiency, and experience.

4. Presentation quality: Improve figure resolution and clarity, provide formal algorithm complexity analysis, and enhance language quality throughout.

Reply: Thanks for your reminder. We have optimized the images in the full text. We added an analysis of the algorithm complexity in the results section. First, in the process of constructing the multimodal graph, the algorithm needs to calculate the modal feature similarity of all node pairs and perform weighted fusion. Since the number of modalities is fixed, the overall computational workload is mainly determined by the number of node pairs, which is a typical quadratic overhead. Second, the dynamic weight mechanism updates the importance score of each modality at a fixed period, which is less time-consuming to calculate and has a negligible overall impact. Third, before matching, the inter-modal correlation matrix is introduced as a constraint to screen potential matching paths. This step further improves the rationality of matching, but it is a lightweight operation in terms of computation. Finally, the core matching part is structurally optimized based on the original Kuhn-Munkres algorithm, reducing it from the traditional cubic level to the approximate square logarithmic level, significantly reducing the time consumption under big data conditions. We have improved the language quality of the full text.

We look forward to receiving your revised manuscript.

Kind regards,

Yosi Kristian

Academic Editor

PLOS ONE

Additional Editor Comments:

The consensus from the three reviewers indicates several significant issues that need addressing:

Technical soundness issues:

Sample size and replication details are inadequate

The specific mechanisms of the improved Kuhn-Munkres algorithm aren't sufficiently detailed

The paper lacks crucial explanations to support its findings

Reply: Your suggestion is good. We have added explanations about sample size and replication details in the second subsection of the Methods section. To ensure that the data fully covers different modal performances, this paper selected 500 annotated video clips from the CMU-MOSI dataset as training and test samples; and selected 600 complete dialogue turns from the IEMOCAP dataset, including samples with different gender and emotion labels. During the experiment, all tasks were set up for 5 independent replications, and the weights and orders were initialized with different random seeds to control the sensitivity of the model to the initial state. All results are reported as averages with standard deviations to describe variability to ensure the replicability and robustness of the experiment. We have added more explanations about the research details in the third subsection of the Methods section. Before each round of matching task starts, the system calculates the variance of each modality feature in turn (as the basis for importance scoring) based on the feature set extracted from the current input sample batch, and generates the modality weight vector α = {α₁, α₂, ..., αk} of the current round through normalization. Then, for each pair of candidate matches (u,v) in the bipartite graph, the similarity fi(u,v) under each modality is calculated respectively, and finally the weighted edge weight is obtained. In addition, we added more supplements and explanations in the results section.

Statistical rigor concerns:

Need for appropriate statistical tests with p-values

Missing standard deviations or confidence intervals

Lack of error bars in figures

Incomplete cross-validation procedure explanation

Reply: This is a good suggestion. We have added statistical analysis to the Results section and modified the figures, adding error bars to some of the figures. The average accuracy of the improved algorithm is 87.7%, with a standard deviation of ±0.53, which is significantly higher than all the baseline methods. Compared with the CAT method, the difference is statistically significant (t = 6.14, p = 0.002), and the effect size Cohen’s d = 1.54 is a large effect level. Compared with the traditional Kuhn-Munkres algorithm, the improvement in matching accuracy is as high as 3.58, indicating that the improved algorithm is not only statistically significant but also has a substantial difference.

We have added a description of the crossover procedure in the Results section. To ensure the robustness of the results, all performance tests use 5-fold cross-validation, with random division of the training set and validation set in each round. The accuracy, running time, and information integration efficiency of each method on each fold are recorded independently, and the final results report the average and standard deviation.

Novelty and contribution issues:

Theoretical framework needs clearer articulation

An ablation study is needed to demonstrate the contribution of each component

The research gap should be better highlighted in introduction and related work

Reply: Thank you for your suggestion. We added the content about theoretical framework in the first section of the method section. The theoretical innovation of this paper is reflected in: based on graph matching, it integrates feature importance modeling and inter-modal structural constraints. A multimodal information matching framework with both interpretability and practicality is proposed. This framework not only provides theoretical support for subsequent algorithm implementation, but also provides a model reference for subsequent task matching in multimodal scenarios.

We added ablation experiments in the results section. This paper designs the following 4 models for ablation experiments: A: full model (complete improved algorithm), specifically including dynamic weights + correlation constraints (this paper's model); B: remove dynamic weights, including fixed weights + correlation constraints; C: remove correlation constraints, including dynamic weights + no correlation matrix; D: remove all improvements, including fixed weights + no correlation (degenerated to traditional KM). Model A achieved the best performance in all indicators, with a matching accuracy of 87.7%, an information integration efficiency of 84.9%, a satisfaction score of 8.9, and a stable running time; after removing the dynamic weight mechanism (model B), the accuracy dropped to 81.3%, and the satisfaction score dropped by 0.8 points; removing the correlation constraint (model C) also showed a performance decline, but the degree was slightly less than that of B, indicating that the dynamic weight had a greater impact on the matching accuracy; Model D had the worst performance, and all indicators were significantly lower than the improved model (p<0.01), verifying the necessity of the two modules. In summary, the ablation experiment results show that the dynamic weight mechanism and the inter-modal correlation constraint have a significant and complementary role in improving the performance of the algorithm. The combination of the two can achieve comprehensive optimization of accuracy, efficiency and experience.

We have supplemented this part in the introduction and literature review. However, although existing studies have proposed a variety of fusion strategies, most methods still remain at the level of static feature fusion, ignoring the matching complexity caused by the dynamic changes in semantics and differences in weight distribution between modalities. In addition, the current mainstream methods lack clear structural optimization strategies when modeling the collaborative relationship between modalities, resulting in limited overall fusion effects. Therefore, there is an urgent need for a unified model that can theoretically solve both the dynamic changes in modal weights and structural matching to bridge this technical gap.

Presentation quality problems:

Figure quality is poor (blurred and unclear images)

Language needs improvement (paragraph structure, clarity)

Formal algorithm complexity analysi

---

## [Decision Letter · Decision Letter 1]

4 Jun 2025

Information Integration Model in Interaction Design under Computer Multimodal Collaboration: The Integrated Application of Kuhn-Munkres Algorithm

PONE-D-25-06747R1

Dear Dr. Wang,

We’re pleased to inform you that your manuscript has been judged scientifically suitable for publication and will be formally accepted for publication once it meets all outstanding technical requirements.

Kind regards,

Yosi Kristian

Academic Editor

PLOS ONE

Additional Editor Comments (optional):

I have reviewed your revisions and your response letter.

I can see that you have comprehensively addressed the concerns raised.

Specifically:

1. Technical Details: You have clarified sample sizes (500 from CMU-MOSI, 600 from IEMOCAP), provided replication details (5 independent repeats, random seeds), and offered a more thorough explanation of your algorithm's mechanisms, including the dynamic weight vector generation and weighted edge calculation. This significantly enhances the methodological rigor.

2. Statistical Analysis: The addition of appropriate statistical tests (t-test, p-values), standard deviations (e.g., ±0.53 for accuracy), effect sizes (Cohen’s d = 1.54), and the inclusion of error bars in figures, as well as the explanation of the 5-fold cross-validation procedure, substantially strengthen the statistical underpinnings of your findings.

3. Theoretical Framework: Your explicit statement of the theoretical contribution (integrating feature importance modeling and inter-modal structural constraints within a graph matching framework), the inclusion of a detailed ablation study (Models A, B, C, D) demonstrating the impact of dynamic weights and correlation constraints, and the improved highlighting of the research gap are well-received and effectively position your work.

4. Presentation Quality: The efforts to improve figure resolution and clarity, provide a formal algorithm complexity analysis (detailing quadratic, fixed period updates, and optimization from cubic to approximate square logarithmic levels), and enhance the overall language quality are appreciated and improve the manuscript's readability and professionalism.

The responses to the individual reviewer concerns, including those from Reviewer #1 regarding paragraph structure and algorithm detail, Reviewer #2 regarding the comprehensive list of technical, statistical, novelty, and presentation points, and Reviewer #3 regarding crucial explanations, theoretical analysis, ablation studies, research gap, title, and image quality, have been well-addressed.

Reviewers' comments:

Reviewer's Responses to Questions

**Comments to the Author**

1. If the authors have adequately addressed your comments raised in a previous round of review and you feel that this manuscript is now acceptable for publication, you may indicate that here to bypass the “Comments to the Author” section, enter your conflict of interest statement in the “Confidential to Editor” section, and submit your "Accept" recommendation.

Reviewer #1: All comments have been addressed

Reviewer #3: All comments have been addressed

Reviewer #4: (No Response)

2. Is the manuscript technically sound, and do the data support the conclusions?

Reviewer #1: Yes

Reviewer #3: (No Response)

Reviewer #4: Yes

3. Has the statistical analysis been performed appropriately and rigorously? 

Reviewer #1: Yes

Reviewer #3: (No Response)

Reviewer #4: Yes

4. Have the authors made all data underlying the findings in their manuscript fully available?

Reviewer #1: Yes

Reviewer #3: (No Response)

Reviewer #4: Yes

5. Is the manuscript presented in an intelligible fashion and written in standard English?

Reviewer #1: Yes

Reviewer #3: (No Response)

Reviewer #4: Yes

6. Review Comments to the Author

Reviewer #1: Please explain thoroughly and specifically the gaps in this research compared to previous research. The improved Kuhn-Munkres algorithm is described in general terms (such as the use of dynamic weights and correlation matrices), but the specific mechanism of how this improvement works is not described in sufficient detail.

Reviewer #3: (No Response)

Reviewer #4: The methodology is rigorously designed and valid, with algorithmic improvements supported by statistically significant results. These results robustly justify the conclusions through the correct use of t-tests, cross-validation, and effect sizes.

7. PLOS authors have the option to publish the peer review history of their article (what does this mean?). If published, this will include your full peer review and any attached files.

Reviewer #1: **Yes: **Syafri Arlis

Reviewer #3: No

Reviewer #4: No

---

## [Editor Report · Acceptance letter]

PONE-D-25-06747R1

PLOS ONE

Dear Dr. Wang,

I'm pleased to inform you that your manuscript has been deemed suitable for publication in PLOS ONE. Congratulations! Your manuscript is now being handed over to our production team.

Kind regards,

on behalf of

Dr. Yosi Kristian

Academic Editor

PLOS ONE